# Exposure to Violence during Childhood and Child-to-Parent Violence: The Mediating Role of Moral Disengagement

**DOI:** 10.3390/healthcare11101402

**Published:** 2023-05-12

**Authors:** Nazaret Bautista-Aranda, Lourdes Contreras, M. Carmen Cano-Lozano

**Affiliations:** Department of Psychology, University of Jaén, 23071 Jaén, Spain

**Keywords:** child-to-parent violence, exposure to violence, moral disengagement, adolescents, mediator

## Abstract

This study examines the influence of exposure to family violence during childhood on child-to-parent violence (CPV) through moral disengagement. The sample included 1868 Spanish adolescents aged between 13 and 18 years (57.9% female, *M_age_* = 14.94, *SD* = 1.37). Participants completed the Child-to-Parent Violence Questionnaire, the Mechanisms of Moral Disengagement Scale, and the Exposure to Violence Scale during childhood. Results showed that exposure to family violence during childhood (vicarious and direct violence) contribute independently and positively to CPV. Moreover, the relationship between vicarious and direct exposure to family violence and CPV is mediated by moral disengagement. The structural model was replicated for both CPV towards the father and CPV towards the mother. The results highlight the importance of early exposure to family violence and moral disengagement in violent behavior towards parents. It is necessary to stage an early intervention with children who have been exposed to family violence in order to prevent an intergenerational transmission of violent behaviors.

## 1. Introduction

Child-to-parent violence (CPV) is a type of family violence initially defined by Harbin and Madden [1] who coined the term “Battered Parent Syndrome” which exclusively included physical aggression and verbal/non-verbal threats of physical harm. The definition has evolved and currently refers to any act of physical, psychological, or financial abuse by children to parents through which children obtain power and control over the parents [2]. Recently, it has added to the concept that this type of violence is carried out consciously, intentionally, and repeatedly over time to dominate and coerce parents (or persons who take their role) [3,4].

In recent years, a great deal of research has examined the phenomenon of CPV, analyzing the different individual, familial, and societal factors related to the origin and maintenance of CPV (see the review by Simmons et al. [5]). Given that this type of violence is displayed in the family setting, special interest has been shown in variables related to family structure and dynamics [6,7,8,9,10,11]. One of the most empirically supported variables is exposure to family violence. Two forms of exposure to violence are distinguished, a vicarious one, when children witness violence between parents, and a direct one, when children are victims of violence by their parents. Social learning theory [12] suggests that children from violent homes may acquire aggressive patterns of responses from observational learning and reinforcement of aggressive adult models, and, consequently, use violence as a solution to deal with interpersonal conflicts. Thus, CPV could occur as a reaction to a previous violent experience or as an acquired response through social learning. The results of numerous studies on CPV are consistent with this approach, finding that a large proportion of adolescents who abuse their parents have been exposed to family violence (see the review by Gallego et al. [13]). Specifically, a recent study with a sample of 1559 adolescents from a community population found that more than half of the adolescents who perpetrate CPV had experienced some type of violence within the family (63.4%) [14]. Similar results were found in another study with a judicial sample, according to which 54% of adolescents had witnessed family violence and 25% had suffered direct victimization by their parents [15]. Regarding the role of adolescent gender, Izaguirre and Calvete [16] found that, while daughters are more likely to experience psychological victimization, sons are more likely to experience physical victimization. In turn, the results of numerous investigations have found a positive relationship between exposure to family violence and CPV [17,18,19], and, in particular, between CPV and direct family victimization [6,20,21,22,23,24] and between CPV and vicarious family victimization [20,22,24,25]. Moreover, these results have been found in prospective longitudinal studies [26,27]. In general, it has been shown that CPV is a lagger effect of exposure to violence at home [13]. However, scarce research has delineated the temporal range of exposure to violence during childhood. It is imperative to clearly determine the moment of early exposure to violence in order to differentiate the distant effects from the immediate effects.

While there is considerable consensus regarding the relationship between exposure to violence and CPV, not all children who grow up in violent family environments will inevitably become abusers within the family context in the future [17,18]. Therefore, we understand that exposure to family violence, as a risk factor, does not explain by itself the violent behavior that adolescents perpetrate towards their parents, so it would be interesting to know what variables may influence this relationship. According to social learning theory [12] and the social information processing model [28,29,30,31], past experiences along with personal dispositions and socio-cultural context lead individuals to develop their own knowledge about the world around them, which consequently will guide their information processing in future social interactions. Information processing, in interaction with values and moral norms of behavior [31,32], will determine the effect of life experiences on later behavior. When confronted with a given social situation, the individual assimilates information through a sequence of cognitive steps, such as encoding and interpreting environmental cues, accessing and evaluating a behavioral response in accordance with moral standards, and, finally, enacting a specific response.

Research has examined the relationship between CPV and socio-cognitive variables. For example, Calvete et al. [26] found that justification of violence correlated significantly with CPV. In this line, Orue et al. [33] found that aggressive response access and justification of violence predicted both CPV towards the father and the mother, while anger only predicted CPV towards the mother. On the other hand, Contreras and Cano-Lozano [34], in their study with a judicial sample, found that adolescents who had committed CPV offences presented a higher hostile social perception and a lower ability to anticipate and understand consequences of social behaviors in comparison with other young offenders and non-offenders. Moreover, they found that exposure to violence at home was related to adolescents’ hostile social perception [17]. In the same vein, Simmons et al. [35] found that participants who abused their parents reported higher levels of family violence, as well as different impelling psychological factors of aggression such as trait anger, aggressive scripts, rumination, and impulsivity in the face of negative emotions, but only exposure to family violence and trait anger significantly predicted CPV towards the father and mother. Further investigating this question, Contreras et al. [18] performed a study with a large sample of adolescents from a community population and analyzed the role of different dysfunctional socio-cognitive processing components in the relationship between exposure to violence at home and CPV. Specifically, they found that exposure to violence, including vicarious and direct victimization, was positively related to hostile attribution, anger, aggressive response access, anticipation of positive consequences of aggression, and justification of violence. In addition, anger and aggressive response access were positively related to CPV motivated by reactive reasons, while anticipation of positive consequences of aggression and justification of violence were positively related to the instrumental use of violence towards parents. These results suggest that exposure to violence at home influences the development of maladaptive social-cognitive processing in adolescents which, in turn, will influence the later development of CPV. These adolescents may have learned, through exposure to aggressive adult models, that violence is a legitimate form of resolving conflicts with their parents or obtaining what they desired. In the same line, Cano-Lozano et al. [21] recently reported, after analyzing a sample of young offenders, that direct victimization at home was also related to CPV through social-cognitive processing. Finally, Junco-Guerrero et al. [36] analyzed the effects of exposition to violence within the family and the justification of violence on CPV in a sample of adolescents, in which it was shown that the justification of violence totally mediated, on the one hand, the relationship between being a violence witness and CPV towards mother, and, on the other hand, the relationship between being a violence victim and CPV towards father. Given the relevant role of socio-cognitive variables in CPV, it would be interesting to examine the role of other such variables in the relationship between early exposure to family violence and CPV.

A socio-cognitive variable of special interest, because of its association with violent and antisocial behavior, is moral disengagement. Bandura [32,37] analyzed moral disengagement as a self-regulatory cognitive process between moral standards and moral behavior. In the course of socialization processes, people adopt social norms—standards of social right and wrong—to guide behavior. Acting in line with internal standards provides satisfaction and avoids self-sanctions [32]. However, on occasion, people act contrary to their norms, producing a dissonance between what is thought and what is done. In these situations, self-sanctions are deactivated through moral disengagement, which “refers to the use of different legitimation mechanisms conducive to a selective disengagement of moral censure” [38] (p. 134). Consequently, when self-sanctions—moral censure—are deactivated, violent behavior and harmful acts against others are disinhibited. Moral disengagement can develop in various periods of life. Bandura et al. [39] showed that moral disengagement is present even in the earlier years, especially if the child has been exposed to models and situations of violence. However, moral disconnection is amplified and consolidated during adolescence and young adulthood, when people are experiencing significant changes in their identity, independence, and social relationships [37,40]. Gender differences in moral disengagement do not exist in the earlier years, but later, males are more prone to moral disengagement than females [40]. Bandura [32,37] identifies eight mechanisms of moral disengagement which, in turn, are grouped into four categories at which moral self-sanctions are disengaged: (1) Reconstruction of immoral behavior is the first category which includes three mechanisms, moral justification, euphemistic labeling, and advantageous comparison, which are used to reinterpret prejudicial behavior by making it personally and socially acceptable; (2) Obscuring personal responsibility is the second category which includes two mechanisms, diffusion of responsibility and displacement of responsibility, both of which are used to obscure and minimize one’s own responsibility for the prejudicial behavior; (3) Misrepresenting injurious consequences is the third category in which the mechanism distortion of consequences would act to distort the harmful consequences of the abusive behavior; (4) Blaming the victim is the fourth and last category which includes two mechanisms, dehumanization and attribution of blame, which would facilitate the depersonalization and blaming of the victim.

Numerous investigations have observed that moral disengagement is positively associated with aggression and violence in children and adolescents [41,42,43,44] and, in contrast, it has been negatively related to prosocial behaviors [41,45,46]. Regarding violent behavior in adolescence, moral disengagement is positively related to dating violence [47,48] and also to bullying and cyberbullying [38,49]. However, to date, there are no studies that have examined moral disengagement in the context of CPV, so it would be interesting to explore this relationship.

Most studies have analyzed the role of moral disengagement in relation to aggressive and antisocial behaviors as a self-regulation process [39,41,44] according to which people selectively deactivate internal moral standards for their own actions and thus facilitate the performance of aggressive behaviors [32,37], so, from this approach, moral disengagement could act as a mediating variable. Some empirical studies have shown that moral disengagement mediates the relationship between different family variables, such as parental attachment, positive parenting and rejecting parenting, and aggressive and delinquent behavior [50,51,52]. Some studies have also examined the relationship between moral disengagement and exposure to violence. For example, Wojciechowski [53], in a longitudinal study using a sample of juvenile offenders, found that polyvictimization was related to violent offending, through moral disengagement and lack of impulse control. Moreover, other studies have observed that moral disengagement mediated the effect of exposure to family violence on bullying and cyberbullying [54,55,56,57]. However, there are no studies that examine the mediating role of moral disengagement in the relationship between early exposure to family violence and CPV.

In short, the scientific literature highlights the importance of the violent environment in which children and adolescents experience violence for the development and maintenance of CPV, although few studies have examined in depth how family violence is associated with CPV through socio-cognitive variables. In addition, numerous studies show that cognitive self-regulation processes of behavior (e.g., moral disengagement) promote aggressive behavior in adolescence, but moral disengagement have not been studied in relation with the phenomenon of CPV. For these reasons, the present study aimed to examine the role of moral disengagement in the relationship between early exposure to family violence and CPV. Specifically, the following objectives were proposed. The first objective was to analyze the relationship between exposure to family violence that occurred before the age of 10 years (exposure to violence between parents and parent-to-child violence) and CPV towards the father and mother. The second objective was to examine the relationship between moral disengagement and CPV towards the father and mother. Finally, the third objective was to analyze the mediating role of moral disengagement in the relationship between exposure to family violence in childhood (vicarious and direct exposure) and CPV towards the father and mother.

Based on previous literature, the following hypotheses were established:

**Hypothesis** **1.***CPV will be significantly and positively related to exposure to violence between parents and to parent-to-child violence during childhood [26,27]*.

**Hypothesis** **2.***Moral disengagement will be significantly and positively related to CPV [41,42,43,44]*.

**Hypothesis** **3.***We expect moral disengagement to mediate the relationship between early exposure to family violence (vicarious and direct exposure, independently) and CPV [53,54,55,56,57]*.

The mediational theoretical model proposed to analyze the mediating role of moral disengagement in the relationship between vicarious and direct exposure to family violence and CPV is presented in Figure 1.

## 2. Materials and Methods

### 2.1. Participants

The sample comprised 1868 Spanish adolescents (57.9% female, 42.1% male) between the ages of 13 and 18 years (*M_age_* = 14.94, *SD* = 1.37). Participants were recruited from 25 secondary schools located in Ciudad Real (59%), Córdoba (26.2%), Granada (9.3%), and Asturias (5.4%) (Spain). Most participants reported that their parents lived together (92.2%), while 7.2% reported that their parents were divorced or separated. Most of the participants indicated that they were biological children of their parents (98.1%). Regarding the number of siblings, 63.9% indicated that they had one sibling, 23.5% reported having two or more siblings, and 12.5% indicated that they were an only child.

Among the participants, 58% reported that the economic situation of the family was medium-sufficient, 10.4% had a high economic situation, while for the rest (3.9%), it was low-sufficient. Finally, 33.9% of fathers and 35.5% of mothers had completed secondary school, 20.6% of fathers and 27.3% of mothers had completed university studies, 33.6% of fathers and 28% of mothers had completed primary school, and 7.3% of fathers and 4.8% of mothers had no formal education.

### 2.2. Instruments

Child-to-Parent Violence Questionnaire, adolescent version (CPV-Q-A [58]): This scale consists of 14 parallel items (14 items for the father, *α* = 0.682; 14 items for the mother, *α* = 0.711) that evaluate different acts of psychological (4 items), physical (3 items), and financial violence (3 items), as well as control and domain behaviors over parents (4 items). Adolescents were asked to indicate how often they have perpetrated each of the behaviors towards their parents during the last year, using a 5-point response scale: 0 (never), 1 (rarely = it has occurred once), 2 (sometimes = 2–3 times), 3 (many times = 4–5 times), and 4 (very often = 6 times or more).

Exposure to Violence Scale (VES [59]), subscale of Exposure to Violence at Home, assesses both direct and vicarious exposure to violence at home: The scale assessed exposure to violence during childhood (before the age of 10 years). An example of an item assessing direct exposure to violence would be “How often has your father hit or physically hurt you at home?”, while for vicarious exposure to violence, an example would be “How often have you seen your father hit or physically hurt your mother?”. The response scale is a 5-point scale: 0 (never), 1 (once), 2 (sometimes), 3 (many times), and 4 (every day). The Cronbach’s *α* was 0.872 for direct exposure to violence and 0.783 for vicarious exposure to violence.

Mechanisms of Moral Disengagement Scale, Spanish version (MMDS-S, [39,60]): This scale measures the inclination to use different mechanisms of moral disengagement: (1) moral justification, (2) euphemistic labeling, (3) advantageous comparison, (4) diffusion of responsibility, (5) displacement of responsibility, (6) distortion of consequences, (7) dehumanization, and (8) attribution of blame. This scale is composed of 32 items, each of which was answered with a 5-point scale: 1 (fully disagree), 2 (disagree more than agree), 3 (neither agree nor disagree), 4 (agree more than disagree), and 5 (fully agree). The Cronbach’s *α* was 0.854.

### 2.3. Procedure

The research started by obtaining a favorable report from the Ethics Committee of the University of Jaén, Spain (Ref. OCT.19/1.PRY). Subsequently, the permission from the Public Administration of Education was obtained, and, also, the directors of the secondary schools, who were invited to participate and received detailed information on the research objectives. While selecting the educational centers, we considered the type of school (public vs. private) and the school year of the adolescents, as well as the gender of the participants. The secondary schools that indicated their interest and willingness to participate in the study provided informed consent to both parents and children. In the case of participants under 18 years of age, both they and their parents had given informed consent. Each participant was assigned an identification code. No incentive was offered for participation; participation was voluntary, anonymous, and confidential. The administration of paper questionnaires was carried out in groups in their classrooms, with each session lasting approximately an hour. The questionnaires were administered by investigators trained to conduct such surveys.

### 2.4. Data Analysis

A significance level of 0.05 was established for all analyses. We performed descriptive analyses including means, standard deviations, skewness, and kurtosis. The possible relationships between the study variables were examined using Spearman correlations, given that the corrected Lilliefors significance of the Kolmogorov–Smirnov test reported the distribution of the data in the study variables did not represent a normal distribution, something commonly observed in studies of violent behaviors in community populations. The Cronbach’s *α* was also calculated to examine the overall reliability of each of the instruments used in this study. Next, hierarchical regression analyses were performed to explore the contribution of the study variables (vicarious and direct exposure to violence and moral disengagement) on CPV. In Model 1, the gender of the participants was introduced as a control variable (gender: 1 = female) as well as the exposure to violence at home. Moral disengagement was included in Model 2. Structural equation modeling (SEM) was performed with STATA v. 16 for testing the mediation hypothesis, in which exposure to family violence increases CPV through moral disengagement. We used the Satorra–Bentler correction as the maximum likelihood estimation method due to the non-compliance with the multivariate normality assumption, given that Mardia multivariate test of skewness and kurtosis were significant [61]. The overall model fit was estimated with conventional indicators: root mean square error approximation (RMSEA), Comparative Fit Index (CFI), Tucker–Lewis Index (TLI), and standardized root mean square residual (SRMR). According to Hu and Bentler [62], obtaining a cutoff value close to or above 0.95 for CFI and TLI, a cutoff value close to or below 0.06 for RMSEA, and a cutoff value close to or below 0.08 for SRMR are needed before concluding that there is a good fit of the model.

## 3. Results

Table 1 presents the means, standard deviations, and correlations between the variables in this study. Both CPV-Father and CPV-Mother were positively and significantly related to vicarious and direct exposure to family violence and moral disengagement.

Table 2 indicates that vicarious and direct exposure to family violence contributed independently and positively to CPV towards the father and towards the mother (Model 1). When moral disengagement was included (Model 2), the contribution of vicarious and direct exposure to family violence to CPV towards both father and mother decreased. These results suggest that moral disengagement may be a mediator between vicarious and direct exposure to family violence and CPV towards the father and towards the mother.

Prior to the mediational model, the direct effect of vicarious and direct exposure to family violence on CPV in the absence of the mediator was evaluated. We found positive and significant direct effects between vicarious exposure to family violence and CPV (CPV-Father, *β* = 0.248, *SE* = 0.064, *p* < 0.001; CPV-Mother, *β* = 0.245, *SE* = 0.060, *p* < 0.001) and also between direct exposure to family violence and CPV (CPV-Father, *β* = 0.301, *SE* = 0.045, *p* < 0.001; CPV-Mother, *β* = 0.315, *SE* = 0.042, *p* < 0.001). The model presented an adequate fit for CPV towards the father [*χ^2^*(20) = 47.159, *p* < 0.01, CFI_SB = 0.980, TLI_SB = 0.963, RMSEA_SB = 0.027, SRMR = 0.029] with 13.5% of the variance explained. There was also an adequate fit for CPV towards the mother [*χ^2^*(20) = 52.315, *p* < 0.001, CFI_SB = 0.977, TLI_SB = 0.959, RMSEA_SB = 0.029, SRMR = 0.030] with 15.6% of the variance explained.

Subsequently, moral disengagement was introduced as a mediator of the relationship between vicarious and direct exposure to family violence and CPV. Figure 2 presents the results of the analysis of the proposed models for CPV towards the father (Figure 2A) and CPV towards the mother (Figure 2B). The indirect effect of vicarious and direct exposure to family violence on CPV through moral disengagement was positive and significant (vicarious exposure: CPV-Father, *β* = 0.037, *SE =* 0.014, *p* < 0.01; CPV-Mother, *β* = 0.039, *SE* = 0.015, *p* < 0.05; direct exposure: CPV-Father, *β* = 0.042, *SE* = 0.009, *p* < 0.001; CPV-Mother, *β* = 0.045, *SE* = 0.0210, *p* < 0.001). The partially mediated model revealed a good fit of the data for CPV to the father [*χ^2^*(107) = 359.051, *p* < 0.001, CFI_SB = 0.953, TLI_SB = 0.940, RMSEA_SB = 0.036, SRMR = 0.033] and explained 17.3% of the variance of CPV to the father. Similarly, the partially mediated model for CPV towards the mother showed an excellent fit [*χ^2^*(107) = 371.692, *p* < 0.001, CFI_SB = 0.952, TLI_SB = 0.939, RMSEA_SB = 0.036, SRMR = 0.034] and explained 19.4% of the total variance for CPV towards the mother. Both models were controlled for the gender variable. The results revealed that males scored higher in moral disengagement, while females scored higher in vicarious and direct exposure to violence at home and CPV.

## 4. Discussion

This study examines the role of moral disengagement in the relationship between early exposure to family violence and CPV. Specifically, the first objective was to analyze the relationship between CPV and vicarious and direct exposure to family violence during childhood. The results confirmed Hypothesis 1. The vicarious exposure to violence and direct exposure to violence at home correlated positively and independently with aggressive behaviors towards parents. In line with previous studies, the results suggest that both exposure to violent behavior employed by parents to relate between themselves [20,22,24,25] and violence with children [6,20,21,22,23,24] promotes the later development of aggressive behaviors of children against their parents. These results support the perspective of intergenerational transmission of violence [63], suggesting that aggressive behaviors are learned by observation and imitation of aggressive adult models of great influence, such as parents [12]. Adolescents who grow up in a violent family environment may normalize, approve, and justify the use of violence as an acceptable response to resolve conflicts with their parents or to obtain what they desire. However, until now, scarce research has delimited the temporal range in which exposure to family violence occurs, and it is difficult to determine the distant or immediate effect that family victimization influences children’s aggressive behavior towards their parents. Our results support the idea that child-to-father violence and child-to-mother violence could be a distant effect in time—a long-term consequence—of the exposure to violence at home suffered in childhood (when children were less than 10 years old). Moreover, the results show that the relationship between CPV is stronger with direct exposure to family violence than with vicarious exposure to family violence, both in the case of fathers and mothers, which is consistent with previous findings [20,22,24]. These results suggest that children who have been exposed to direct violence at home are more likely to perpetrate CPV. In addition, it is observed that violence towards the father and the mother are strongly correlated, which may also indicate that, independently of who exercises violence on the child, the child is likely to perpetrate violence against both parents similarly. 

The second objective of this study was to analyze the relationship between moral disengagement and CPV. The obtained results confirmed Hypothesis 2: moral disengagement correlated significantly and positively with CPV towards the father and towards the mother. These results are in line with previous research that has reported a positive relationship between moral disengagement and aggressive behavior in adolescents [41,42,43,44]. Other studies, in particular, have reported that moral disengagement is present in various forms of violence in adolescence. For example, moral disengagement has been positively related to aggression towards peers at school [38,49] and, also, to aggression against intimate partners in dating [47,48]. Despite the relevance of this variable in aggressive behavior in adolescence, to date, it had not been analyzed in the field of CPV, this being the first study to examine the association between these two variables. The results indicated that moral disengagement is positively associated with father and mother CPV and, therefore, is one of our main contributions. This finding suggests that adolescents who exert violence against their parents present high levels of moral disengagement, which involves decreased self-sanctions for transgressing moral norms and increases the risk of perpetrating violent and detrimental behaviors [12,32], in this case, against their parents.

The third objective of the present study was to analyze the mediating role of moral disengagement in the relationship between vicarious and direct exposure to family violence in childhood and CPV. The data allowed us to confirm the relationship between vicarious and direct exposure to family violence and CPV, and, in addition, to determine the mediating role of moral disengagement as cognitive self-regulation processes of behavior following the approach of Bandura [32,37]. Our results indicated that moral disengagement was a significant factor that positively and partially mediates the relationships between direct family victimization and CPV and between vicarious family victimization and CPV. These results confirmed Hypothesis 3 of this study. Exposure to family violence in childhood in the presence of moral disengagement better predicted father and mother CPV, explaining the 17.3–19.4% result, respectively, compared to 13.5–15.6% that would explain only exposure to family violence. The results are consistent with previous studies that indicate that exposure to violence can increase both moral disengagement and aggressive behavior in adolescents [53,54]. This study adds to the growing empirical evidence that social-cognitive processing plays an important role in adolescents’ aggression towards parents [18,21,33,35]. This study highlights the significant mediating role of moral disengagement in the relationship between family violence suffered during childhood and later violence perpetrated by adolescents towards their parents. This novel finding could indicate that, in cases of CPV, adolescents who were exposed to family violence in childhood may have learned behavioral patterns favorable to violence, internalized beliefs of approval and justification of violence, and normalized violence as a functional response to resolve interpersonal conflicts or get what they want. Therefore, negative past experiences and beliefs of the acceptability of violent behavior could result in the dysfunctional cognitive processing of information and, in turn, increase the dysregulation of moral behavior, thus favoring later aggressive behavior from children to parents. In terms of moral disengagement, adolescents deactivate self-sanctions for immoral behavior performed against parents (e.g., insulting their parents) through moral disengagement mechanisms (e.g., justification of violence, comparing one’s own actions with those of others, disengaging from responsibility for the violent act as well as for the damage caused, or dehumanizing and blaming the victim), thus facilitating the development of CPV.

Finally, with respect to the participants’ gender, we found that males tend to score higher in moral disengagement than females, consistent with the results of previous studies [39,42,46]. Some findings from cross-cultural studies suggest that gender differences in aggressive behavior may reside in the differential tendency of disengaging moral self-sanctions from injurious conduct [40]. In turn, the females scored higher in family violence than males. This finding is consistent with the research, where females, compared with males, report greater exposure to family violence [15]. As for CPV, females show higher levels of CPV than males. Similar results have been found in previous studies [24,36], specifically, it has been found that daughters are more likely to engage in psychological violence while sons tend to engage in physical violence [26].

This study presents some limitations that should be considered when interpreting the results. Firstly, this is a cross-sectional study that precludes the establishment of causal relationships between the variables in this study, so it would be necessary to carry out longitudinal studies to study the long-term effects of exposure to violence in greater depth. Secondly, information is collected in a retrospective form and adolescents are asked to report on what happened in childhood, before the age of ten, leading to recall bias. Thirdly, the sample of this study is made up of adolescents from the Spanish population, particularly from four provinces, which should be taken into account when generalizing the results to other Spanish cities and other countries. Finally, the results are based exclusively on adolescent self-reports, so it would be convenient to have the self-reports of the participants’ own parents.

## 5. Conclusions

This study confirms that vicarious and direct exposure to violence during childhood and moral disengagement both contribute to explaining CPV towards the father and towards the mother. On the one hand, the results indicate that exposure to family violence, both vicarious and direct victimization, in childhood has an important associative and predictive role for CPV, which could indicate that CPV may be a distant effect of exposure to violence at home during childhood. On the other hand, our results provide evidence that moral disengagement is a cognitive mechanism that promotes the violent behavior of adolescents towards their parents. Furthermore, moral disengagement is found to mediate the negative effect of exposure to family violence in childhood on CPV.

The findings have several relevant implications. This study contributes to the understanding of the dysfunctional cognitive processes involved in the relationship between exposure to violence at home during childhood and CPV, also considering gender differences. Moreover, evidence is provided on the relationship between moral disengagement and CPV that, until now, has not been previously examined in the context of CPV. Therefore, the results provided should be considered when designing effective prevention and intervention programs. On the one hand, it is necessary to intervene in order to prevent family violence, so that families should receive training on the negative effects of family violence and adaptive conflict resolution strategies. On the other hand, in cases of CPV, the results suggest the need to intervene with adolescents on the management and relearning of behavioral self-regulation mechanisms (emotional and cognitive). In particular, it is essential to modify beliefs of approval and justification of violence and to promote prosocial and empathic moral reasoning within the family context.

## Figures and Tables

**Figure 1 healthcare-11-01402-f001:**
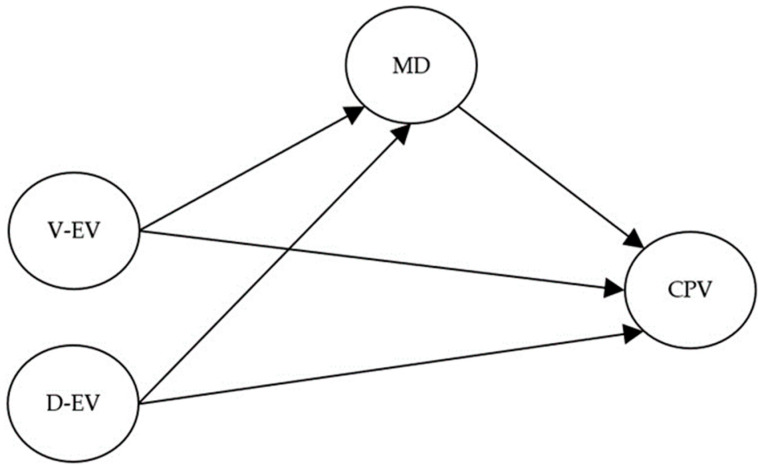
Theoretical mediational model between exposure to violence, moral disengagement, and child-to-parent violence for father and mother. Circles represent latent variables. Dates indicate regressions between the variables. V-EV = vicarious exposure to violence, D-EV = direct exposure to violence, MD = moral disengagement, CPV = child-to-parent violence.

**Figure 2 healthcare-11-01402-f002:**
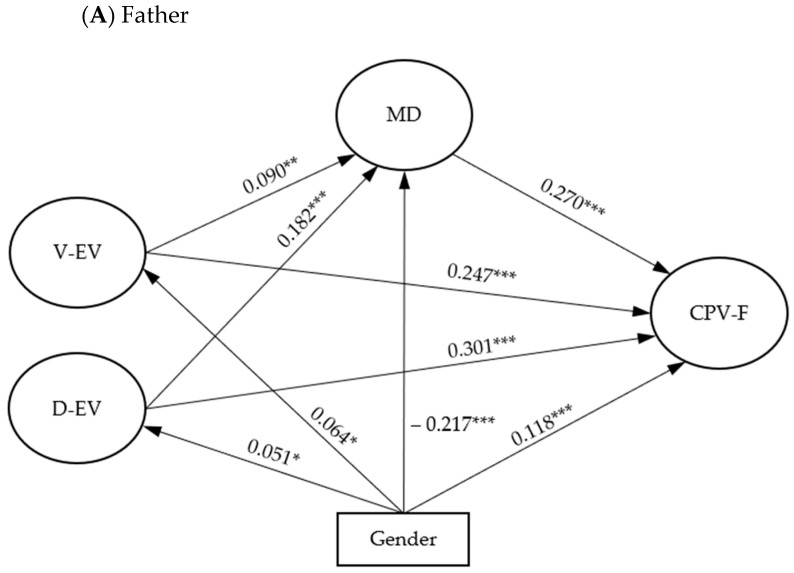
Mediational model for child-to-parent violence. Circles represent latent variables and rectangles represent observed variables. Dates indicate regressions between the variables. V-EV = vicarious exposure to violence, D-EV = direct exposure to violence, MD = moral disengagement, CPV = child-to-parent violence, F = Father, M = Mother. The model for the father is presented in the upper part (**A**), and the model for the mother in the lower part (**B**). * *p* < 0.05, ** *p* < 0.005, *** *p* < 0.001.

**Table 1 healthcare-11-01402-t001:** Means, standard deviations, and Spearman correlations between study variables.

	*M*	*SD*	Skew	Kurtosis	1	2	3	4	5
1. V-EV	0.912	2.205	3.457	14.909	1				
2. D-EV	3.106	4.067	1.313	1.140	0.436 ***	1			
3. MD	66.660	15.880	0.440	0.198	0.133 ***	0.183 ***	1		
4. CPV-Father	5.101	4.227	1.639	4.615	0.235 ***	0.312 ***	0.243 ***	1	
5. CPV-Mother	5.685	4.629	1.885	6.967	0.246 ***	0.344 ***	0.239 ***	0.850 ***	1

Note. V-EV = vicarious exposure to violence, D-EV = direct exposure to violence, MD = moral disengagement, CPV = child-to-parent violence. *** *p* < 0.001.

**Table 2 healthcare-11-01402-t002:** Hierarchical regression analyses for child-to-parent violence as related to early exposure to family violence and moral disengagement.

	Model 1		Model 2	
	*β*	*t*	*β*	*t*
CPV-Father				
Gender	0.053 *	2.444	0.085 ***	3.953
V-EV	0.146 ***	5.986	0.134 ***	5.608
D-EV	0.267 ***	10.922	0.237 ***	9.816
MD			0.201 ***	9.285
*R^2^*	0.135		0.173	
*F*	96.775 ***		97.452 ***	
CPV-Mother				
Gender	0.068 **	3.179	0.100 ***	4.716
V-EV	0.147 ***	6.066	0.134 ***	5.687
D-EV	0.293 ***	12.140	0.263 ***	11.040
MD			0.201 ***	9.406
*R^2^*	0.156		0.194	
*F*	114.652 ***		112.145 ***	

Note. Dependent variables = CPV-Father, CPV-Mother. Gender: 1 = female, V-EV = vicarious exposure to violence, D-EV = direct exposure to violence, MD = moral disengagement, CPV = child-to-parent violence. * *p* < 0.05, ** *p* < 0.005, *** *p* < 0.001.

## Data Availability

The data of this current study are not publicly available due to confidentiality reasons, but are available from the corresponding author upon reasonable request.

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
