# Peer review of "Exposure to Violence during Childhood and Child-to-Parent Violence: The Mediating Role of Moral Disengagement"

_healthcare, 2023, doi:10.3390/healthcare11101402_

Round 1
Reviewer 1 Report
In this article, the author examines the relationship between childhood domestic violence experiences and parental violence against children. The article also examines the underlying mechanisms, such as moral pushback as a mediating mechanism. The authors emphasizes the importance of domestic violence prevention and early intervention for children who have experienced domestic violence to prevent violence from being transmitted from generation to generation. However, the article leaves the following questions to be considered:
1. "There is considerable consensus regarding the relationship between exposure to violence and CPV, establishing a simple association There is considerable consensus regarding the relationship between exposure to violence and CPV, establishing a simple association between both types of violence would be inadequate, as not all children who grow up in violent family environments will inevitably become abusers within Therefore, we understand that exposure to family violence, as a risk factor, does not explain by itself the violent behavior that adolescents perpetrate. Therefore, we understand that exposure to family violence, as a risk factor, does not explain by itself the violent behavior that adolescents perpetrate towards their parents, so it would be interesting to know what variables may influence this relationship. "This discourse aims to emphasize that exposure to violence, by itself, does not explain the violent behavior that children perpetrate toward their parents, so it would be necessary to focus on other variables that influence the relationship. First, it is important to consider whether this is an accurate statement. Second, does this mean that the rest of the paper will focus on the moderating variables along this pathway? Obviously, this is not the purpose of this paper, so it is suggested that this text be modified.
2. What additional variables might have a significant impact on the accuracy of the study results? Is it necessary to consider them in the study design and data analysis?
Author Response
Dear reviewer:
We would like to thank you very much for your comments and suggestions, we think that all of them have contributed to enhance the manuscript. We have followed to each of your comments and, accordingly, we have made different changes on the paper to enhance it. As you will see, we present below a detailed list of the changes we made in response to every comment (in bold letter), following the same sequence:
- "There is considerable consensus regarding the relationship between exposure to violence and CPV, establishing a simple association There is considerable consensus regarding the relationship between exposure to violence and CPV, establishing a simple association between both types of violence would be inadequate, as not all children who grow up in violent family environments will inevitably become abusers within Therefore, we understand that exposure to family violence, as a risk factor, does not explain by itself the violent behavior that adolescents perpetrate. Therefore, we understand that exposure to family violence, as a risk factor, does not explain by itself the violent behavior that adolescents perpetrate towards their parents, so it would be interesting to know what variables may influence this relationship. "This discourse aims to emphasize that exposure to violence, by itself, does not explain the violent behavior that children perpetrate toward their parents, so it would be necessary to focus on other variables that influence the relationship. First, it is important to consider whether this is an accurate statement. Second, does this mean that the rest of the paper will focus on the moderating variables along this pathway? Obviously, this is not the purpose of this paper, so it is suggested that this text be modified. We have modified the text following the recommendation of the reviewer. We have emphasized the relationship between exposure to violence at home and CPV and the relevance of considering other variables that could influence this relationship.
- What additional variables might have a significant impact on the accuracy of the study results? Is it necessary to consider them in the study design and data analysis? We think that additional variable that would have a significant impact would be gender, included in the model. The mediational model presented in this study is simple and parsimonious, exploring the effect of moral disengagement on the relationship between exposure to violence at home and CPV. This is a novel approach, which previously has not been studied in CPV, so including other variables could take us away from the purpose of this study.
Reviewer 2 Report
1. A very interesting and thorough introduction/review of the literature. It may be useful to include a sentence or two on what we know about when (over the lifecourse) moral engagement begins to develop, since the paper aims to draw conclusions around placing intervention for violence prevention in the early years.
2. Could the authors justify the age cut-off for early childhood violence - below age 10 - is it theoretical, developmental, or based on availability of data. I see that the age cut-off is based on the scale used but it might be helpful to be explicit since age categories across childhood are pretty well defined.
3. I would be cautious to refer to the temporal effect of violence exposure in the first ten years of life on CPV in the second decade, as a long-term consequence. Especially if we are not measuring or controlling for violent or aggressive behaviour from the child before age 10, which could exist. I would imagine violence perpetrated in adulthood as a long-term consequence. This does not take away from the findings, but I suggest temporal conclusions are restricted to the intergenerational nature of violence transmission. Again, some discussion on when moral disengagement develops could more clearly establish the temporal effect.
4. There are two very interesting findings that aren't in the discussion - a gentle suggestion to consider adding. The first is that CPV against mother and CPV against father are highly correlated. This strengthens the argument in that the association is strong enough that, despite what we could reasonably expect as some parents with different parenting styles, both parents have similar experiences. This could also be interrogated further as to whether boys or girls perpetrate equal amounts (and types) of violence towards mothers and fathers, respectively.
The second interesting finding that could be elaborated on in line 380 which deals with gender is the interpretation that boys develop higher levels of moral disengagement from lower levels of violence exposure in childhood.
A very interesting and compelling read overall!
Author Response
Dear reviewer:
We would like to thank you very much for your comments and suggestions, we think that all of them have contributed to enhance the manuscript. We have followed to each of your comments and, accordingly, we have made different changes on the paper to enhance it. As you will see, we present below a detailed list of the changes we made in response to every comment (in bold letter), following the same sequence:
- A very interesting and thorough introduction/review of the literature. It may be useful to include a sentence or two on what we know about when (over the life course) moral engagement begins to develop, since the paper aims to draw conclusions around placing intervention for violence prevention in the early years. Following this commentary, a few phrases have been included (lines 129-135), indicating that moral disconnection can develop in the early years of life, although it tends to consolidate during adolescence and adulthood.
- Could the authors justify the age cut-off for early childhood violence - below age 10 - is it theoretical, developmental, or based on availability of data. I see that the age cut-off is based on the scale used but it might be helpful to be explicit since age categories across childhood are pretty well defined. Different theoretical approaches define the period of childhood as generally extending up to 12 years of age. In our study, the age of the participants ranged between 13 and 18 years, and we agreed to reduce the period of childhood to 10 years to make a clear distinction between the childhood and adolescent period.
- I would be cautious to refer to the temporal effect of violence exposure in the first ten years of life on CPV in the second decade, as a long-term consequence. Especially if we are not measuring or controlling for violent or aggressive behaviour from the child before age 10, which could exist. I would imagine violence perpetrated in adulthood as a long-term consequence. This does not take away from the findings, but I suggest temporal conclusions are restricted to the intergenerational nature of violence transmission. Again, some discussion on when moral disengagement develops could more clearly establish the temporal effect. We have added in the limitations of the study that it is necessary to carry out longitudinal studies to explore the effects of exposure to violence on child to parent violence.
- There are two very interesting findings that aren't in the discussion - a gentle suggestion to consider adding. The first is that CPV against mother and CPV against father are highly correlated. This strengthens the argument in that the association is strong enough that, despite what we could reasonably expect as some parents with different parenting styles, both parents have similar experiences. This could also be interrogated further as to whether boys or girls perpetrate equal amounts (and types) of violence towards mothers and fathers, respectively. This is a very interesting comment. We have added information about it at the end of the first paragraph of the discussion, highlighting that independently of who perpetrates violence on the child, it is likely that the child will exhibit violence towards both parents similarly.
- The second interesting finding that could be elaborated on in line 380 which deals with gender is the interpretation that boys develop higher levels of moral disengagement from lower levels of violence exposure in childhood. This is a great commentary. We have included information on gender differences for the variables in this study.
Reviewer 3 Report
Thank you very much for inviting me to read the article. I am not an expert on violence behavior, but it was a pleasure to read the article. Nevertheless, I think that it can improve in some ways, specially results, discussion and conclusions. I think that authors can give more information in order to increase its originality and significance of the content.
Lines 1-174: All is very clear and authors give the most important information in few lines. It is a good theoretical frame. Although, some idees authors give after should be include here. For example, all we just know in relation to gender differences (theories).
Lines 183-257: Materials and Methods. All the information is given in very clear way. But after reading that, we wait for more complete results.
It seems that authors have a lot of information about situation of parents/family (living together or divorced/separated; number of siblings, economic situation of the family,..), but they do not use that to do analyses (variables). There are few sentences dedicated to talk about gender differences, but it does not seem enough. In addition, article distinguishes between psychological, physical and financial violence, and it does not give any results related.
Authors explain that their hypothesis are confirmed and, as a result, the study give little new information about “child-to-parent violence”. In general terms, the results are consistent with precious findings. And why did authors though that the mediating role of moral disengagement could be different? Why is studying this important?
The results are very general and give little information, especially if we take into account the methodology. Authors can:
a) Give more information in the way they think it is possible (differences according to economic situation, number of siblings,...; differences violence to mother / father; psychological, physical and financial violence,...)
b) Or, at least, it would be necessary to give more information about moral disengagement. Did authors really think they would get another result?
Line 385-393: When authors talk about limitations, it seems that they feel it is not a good work. I suggest to review that.
a) Causal relationships between the variables. I think that it would be better to talk about correlations. But, in any way, I do not think it can be considered a limitation, because it is determinated by methodology choseen.
b) Memory. Social research can be about memories. It is not a limitation, it is a methodological option. Sample is formed by adolescents and the authors want to know the influence of the past in the present.
c) Results can not be generalizing to other Spanish cities and it is not necessary. Study is about the influence of the exposure to family violence during childhood. Perhaps we should talk about cultural roots, but not differences according to cities/countries.
d) Based exclusively on adolescent self-reports. Why authors think they need also self-reports of the parents? I do not think it is necessary or it could be another research (comparative research)
Author Response
Dear reviewer:
We would like to thank you very much for your comments and suggestions, we think that all of them have contributed to enhance the manuscript. We have followed to each of your comments and, accordingly, we have made different changes on the paper to enhance it. As you will see, we present below a detailed list of the changes we made in response to every comment (in bold letter), following the same sequence:
1. Lines 1-174: All is very clear and authors give the most important information in few lines. It is a good theoretical frame. Although, some idees authors give after should be include here. For example, all we just know in relation to gender differences (theories). Following this commentary, we have included information on gender differences for the variables in this study (lines 52-55; 134-135; 494-504).
2. Lines 183-257: Materials and Methods. All the information is given in very clear way. But after reading that, we wait for more complete results.
- It seems that authors have a lot of information about situation of parents/family (living together or divorced/separated; number of siblings, economic situation of the family,..), but they do not use that to do analyses (variables). There are few sentences dedicated to talk about gender differences, but it does not seem enough. In addition, article distinguishes between psychological, physical and financial violence, and it does not give any results related. The aim is to explore the effect of moral disengagement on the relationship between exposure to family violence and CPV. We have provided this information to describe the sample, so including these variables would take us away from the purpose of this study.
- Authors explain that their hypothesis are confirmed and, as a result, the study give little new information about “child-to-parent violence”. In general terms, the results are consistent with precious findings. And why did authors though that the mediating role of moral disengagement could be different? Why is studying this important? Moral disengagement has been analyzed in other forms of violence except in CPV. In addition, other studies have shown that cognitive variables mediate the relationship between exposure to violence and CPV; however, moral disengagement has not been considered a mediating variable in this relationship. We think that it is necessary to empirically demonstrate the relationship between the study variables, in line with the already existing theoretical and empirical literature.
3. The results are very general and give little information, especially if we take into account the methodology. Authors can:
- Give more information in the way they think it is possible (differences according to economic situation, number of siblings,...; differences violence to mother / father; psychological, physical and financial violence,...). This information is interesting. However, if we include this information, we would be moving away from the objective of this study.
- Or, at least, it would be necessary to give more information about moral disengagement. Did authors really think they would get another result? Based on the theoretical and empirical literature about moral disengagement and aggression, we hypothesize that moral disengagement would be related to CPV and that it would also mediate the relationship between exposure to violence at home and CPV. Until now, moral disengagement in CPV has not been studied, and we provide empirical results on this question.
4. Line 385-393: When authors talk about limitations, it seems that they feel it is not a good work. I suggest to review that.
- Causal relationships between the variables. I think that it would be better to talk about correlations. But, in any way, I do not think it can be considered a limitation, because it is determinated by methodology choseen.
- Social research can be about memories. It is not a limitation, it is a methodological option. Sample is formed by adolescents and the authors want to know the influence of the past in the present.
- Results cannot be generalizing to other Spanish cities and it is not necessary. Study is about the influence of the exposure to family violence during childhood. Perhaps we should talk about cultural roots, but not differences according to cities/countries.
- Based exclusively on adolescent self-reports. Why authors think they need also self-reports of the parents? I do not think it is necessary or it could be another research (comparative research)
We consider that the limitations described in this study are necessary to be cautious when interpreting the results and, in addition, we raise questions to improve future research. In the conclusions section of this study, the strengths are presented.
Reviewer 4 Report
The current study addresses an interesting social topic. The introduction adequately discusses the theoretical aspects of the study, the objectives, and the hypotheses. However, the issue of the control variable should be addressed better. However, other control variables should be included in the analysis to identify the risk of family violence better.
Data analysis. Please specify why it was performed the Spearman correlation.
Please, provide information about the properties of the data set in terms of univariate and multivariate distribution of the data as well as potential outliers and, eventually, the strategy adopted to handle them.
Please, move Figure 1 to the first part of the text.
Results. Please include in Table 1 the values of Kurtosis and skewness.
Please, specify in Table 2 the name of the dependent variable.
I think it could be adequate to run a single model and control the analysis for gender.
Since the study is cross-sectional, testing an alternative model could improve its impact.
Author Response
Dear reviewer:
We would like to thank you very much for your comments and suggestions, we think that all of them have contributed to enhance the manuscript. We have followed to each of your comments and, accordingly, we have made different changes on the paper to enhance it. As you will see, we present below a detailed list of the changes we made in response to every comment (in bold letter), following the same sequence:
- The current study addresses an interesting social topic. The introduction adequately discusses the theoretical aspects of the study, the objectives, and the hypotheses. However, the issue of the control variable should be addressed better. However, other control variables should be included in the analysis to identify the risk of family violence better. We think that additional variable that would have a significant impact would be gender, included in the model. The mediational model presented in this study is simple and parsimonious, exploring the effect of moral disengagement on the relationship between exposure to violence at home and CPV. This is a novel approach, which previously has not been studied in CPV, so including other variables could take us away from the purpose of this study.
- Data analysis. Please specify why it was performed the Spearman correlation. We include a sentence to clarify why we used Spearman correlations to know the possible relationships between the variables in this study. We used Spearman correlations because the distribution of the sample in the study variables did not represent a normal distribution, something commonly observed in studies of violent behaviors in community populations.
- Please, provide information about the properties of the data set in terms of univariate and multivariate distribution of the data as well as potential outliers and, eventually, the strategy adopted to handle them. Information on the univariate and multivariate distribution of the data is included. For the univariate distribution, the Kolmogorov-Smirnow test was performed. In the case of the multivariate distribution, the Mardia multivariate test of skewness and kurtosis. Regarding outliers, we have opted to maintain the outliers in the statistical analysis. It is common in studies on violence with community samples to find outliers or extreme cases outside the standard. These cases can be important for understanding the nature and magnitude of the violence problem in this population, and their elimination could reduce the quality and relevance of the study.
- Please, move Figure 1 to the first part of the text. Corrected.
- Please include in Table 1 the values of Kurtosis and skewness. We have included skewness and kurtosis in Table 1. Also, it has been referred to in the data analysis section.
- Please, specify in Table 2 the name of the dependent variable. Corrected.
- I think it could be adequate to run a single model and control the analysis for gender. The gender control variable refers to the gender of the adolescents. The Child-to-Parent Violence Questionnaire, adolescent version (CPV-Q-A) assesses violence toward the father and the mother separately; for this reason, two models were made for this study, one toward the father and the other toward the mother. In short, the gender of the adolescents and the parents was taken into account.
- Since the study is cross-sectional, testing an alternative model could improve its impact. Based on the review of previous literature, this model can provide more information about this type of violence. Thus, another type of model would not be supported at a theoretical or empirical level.